# Satisfaction of chronic patients with community-based health insurance schemes and related factors: Explanatory sequential mixed methods study

**Desalew Birhan[1], Zewudie Aderaw[2], Eskeziaw Agdew[1], Melkamu Siferih[ID][3]***

**1** Department of Public Health, College of Medicine and Health Sciences, Debremarkos University, Debremarkos, Amhara Region, Ethiopia, **2** Department of Public Health, St. Paul's Hospital Millennium Medical College, Addis Ababa, Ethiopia, **3** Department of Obstetrics and Gynecology, School of Medicine, Debremarkos University, Debremarkos, Amhara Region, Ethiopia

* siferihmelkamu@gmail.com

**Data Availability Statement:** All relevant data are within the paper.

## Abstract

Chronic disease poses a serious threat to accessible, high-quality healthcare. Community-based health insurance (CBHI) schemes provide the poor with financial security. However, there is no evidence in Ethiopia on how satisfied chronic patients are with the schemes. The objective of the current study was to evaluate the satisfaction of chronic patients with the schemes and identify contributing factors. A hospital-based explanatory sequential mixed methods study on 632 chronic patients and 12 key informants was carried out between February 28 and May 31, 2022, in the hospitals of East Gojjam, Northwest Ethiopia. Hospitals and study participants were selected using multistage sampling methods. Quantitative data was entered using Epi Data 3.1 and exported to SPSS version 25 for analysis. P value <0.05 was used to consider significant association in multivariable binary logistic regression. Thematic analysis was a method to manually review qualitative data. A narrative approach was used for integrating the two data. The mean age of study participants was 46.1 (46.1± 5.2, range: 25–82). Patients aged 45 to 64 made up the majority of the population (60.6%). Rheumatoid arthritis affected the preponderance of individuals (36.4%). The overall level of satisfaction of chronic patients with the CBHI scheme was 31% (95% CI; 27–35%). Respect and friendliness (AOR = 7.05; CI: 3.71–13.36), knowledge of benefits packages (AOR = 2.02; CI: 1.24–3.27), partial or non-availability of drugs (AOR = 0.24, AOR = 0.21, respectively), waiting times (AOR = 1.84; CI: 1.12–3.0), and availability of laboratory tests (AOR = 1.59; CI: 1.01–2.48) were significantly associated with participants' satisfaction with the schemes. Our study revealed that the overall satisfaction of chronic patients was quite low and affected by the availability of drugs and laboratory tests, caregivers' respect and friendliness, waiting times, and participant knowledge. Therefore, stakeholders must concentrate on reducing waiting times, improving the availability of drugs and laboratory tests at each hospital, opening back up community pharmacies, and promoting awareness about benefits packages primarily through health education. The main focus of researchers needs to be on nationally representative studies that include more important factors.

**Funding:** The authors received no specific funding for this work.

**Competing interests:** The authors have declared that no competing interests exist.

## Introduction

Chronic diseases such as chronic hypertension, diabetes, epilepsy, and others are now much more widespread than they were in the past [1, 2]. They were the main factors in Ethiopia's age-standardized death rate, accounting for 711 deaths per year per 100,000 inhabitants, similar to the majority of low-income nations [3].

It has become more challenging for the healthcare system, which is financed by out-of-pocket payments, due to the rising expenditures of chronic diseases [2]. In Ethiopia, where out-of-pocket costs were projected to be 18.2 billion ETB in 2015–16 while chronic disease treatment contributed to 23% of all out-of-pocket costs, the financial strain of chronic diseases is compounded [4]. The nation's response to chronic illnesses is still scattered, and the amount spent on chronic illnesses per person is still negligible [5].

Community-based health insurance (CBHI) schemes are regarded as one viable strategy to reduce out-of-pocket costs and attain universal health coverage in low-income nations [6–8]. Particularly, the CBHI is a very helpful tool for lowering the cost of health care for those who cannot afford it to receive treatment for chronic diseases [9, 10]. Ethiopia is dedicated to achieving universal health coverage (UHC) through expanding high-quality healthcare services that are available, affordable, and acceptable to all households experiencing poverty owing to high out-of-pocket costs. The country has started implementing a comprehensive and long-lasting financial risk protection program with a community health insurance (CBHI) scheme because it is an essential element of UHC. The scheme premium is set at the household level and not individuals [11, 12].

CBHI beneficiaries were the subject of the vast majority of studies on CBHI schemes [13–20]. However, it is ideal to incorporate the provider-side constraints. Few qualitative and mixed-method studies have tried to identify the reasons for chronic patients' dissatisfaction with CBHI, despite the enormous burden of chronic diseases and escalating costs. Little is known about chronic patients, in particular their satisfaction with CBHI schemes in Ethiopia.

Therefore, the current study sought to assess the level of chronic patients' satisfaction with the CBHI schemes, and its contributing factors, and to explore the perspectives of CBHI beneficiaries and healthcare providers at East Gojjam Zone Hospitals in Northwest Ethiopia.

## Methods and materials

### Ethical approval and consent to participate

The Declaration of Helsinki served as a guide for the ethical standards. Ethical clearance was obtained from the Institutional Research Ethics Review Committee of Debremarkos University with reference number: **HSC/RC/C/Ser/CO/80/11/14**. An official letter of permission was obtained from Debremarkos University, the corresponding CBHI agency offices, and the respective East Gojjam zone hospitals. Supportive letters were taken from each hospital administrator. In addition, informed written consent to participate was taken from all study participants including key informants after providing brief explanations about the purpose and procedure of the study. To maintain the confidentiality of collected data, anonymity was maintained throughout the research process. Furthermore; the right to withdraw from participation at any time was respected.

### Study setting, design, and participants

A basic mixed methods study employing a sequential explanatory design was undertaken in the hospitals of the East Gojjam Zone in the Amhara Region. East Gojjam Zone is home to 11 hospitals. Debre Markos Hospital is a Comprehensive Specialized Referral facility, Shegaw

Motta is a general hospital, and the remaining facilities are primary hospitals [21]. There were around 5749 chronic disease follow-up CBHI members among the hospitals that were chosen. As a source population, the CBHI schemes included all adult chronic disease follow-up patients at the hospitals in the East Gojjam Zone.

**For the quantitative section (cross-sectional study)**: the study population consisted of all chronic patients who were household heads, at least 18 years old, were enrolled in the CBHI for at least one month and in the active follow-up in the outpatient department at the particular hospitals. With regard to **the qualitative component**, key informants were chosen from healthcare providers from the outpatient department staff, CBHI agency executive officers, hospital chief executive officers, and once more, carefully selected chronic follow-up patients who were community leaders. Patients with life-threatening illnesses or those who couldn't communicate were excluded.

## Sample size determination and sampling technique

Assuming that 54.1% of household heads with chronic disease follow-up in the outpatient department in each hospital were satisfied with the community-based health insurance schemes, the sample size was determined using a single population proportion formula from the previous study [14] done in Anilemo District, Hadiya Zone, Southern Ethiopia, 95% level of confidence and, 5% degree of precision. The sample size for this study, then, was 382. By taking a 10% non-response rate into account, the necessary minimum sample size would be calculated as = 382+38.2 = 421. Due to the multistage sampling method and design effect used in the study, the ultimate sample size was 421×1.5 = 632.

Maximum variation of the phenomenon under study was used to determine the sample size for qualitative data. Twelve key informants (five chronic follow-up patients, four CBHI executive officers, one hospital chief executive officer, and two health care providers (one nurse, and one medical doctor)) were purposively selected. The first eleven hospitals in the tier system were classified according to their level, and then four hospitals were chosen using a simple random sampling technique. A systematic random sampling technique was used to select the actual study participants. The sampling interval was established by dividing the required sample number for a given day by the daily average of the number of chronic disease follow-ups. Study participants were distributed proportionally every tenth interval using a systematic random selection technique to pick the participants (**Fig 1**).

## Study variables

The **dependent variable** was satisfaction with the CBHI schemes. **Independent variables** were **socio-demographic characteristics** including age, marital status, level of education, occupation, and monthly family income; **enrollee's knowledge of the health insurance benefit package** including good strategy to assist customers, coverage for outpatient care, coverage only for care from public health services, coverage for inpatient care, coverage only for care provided domestically, doesn't cover transportation costs, and coverage for cosmetic values; **CBHI experience and management-related elements** including fee reimbursement, CBHI-related meetings, voluntary enrollment, length of enrollment, card renewals, the type of health facility visited, and community-health insurance membership type. Moreover, **health service-related factors** include waiting times, being treated as CBHI members in the same manner as out-of-pocket patients, respect and friendliness from healthcare providers, the availability of drugs in hospitals, referral systems, queue management procedures, laboratory service provision, and complaint handling systems (officers).

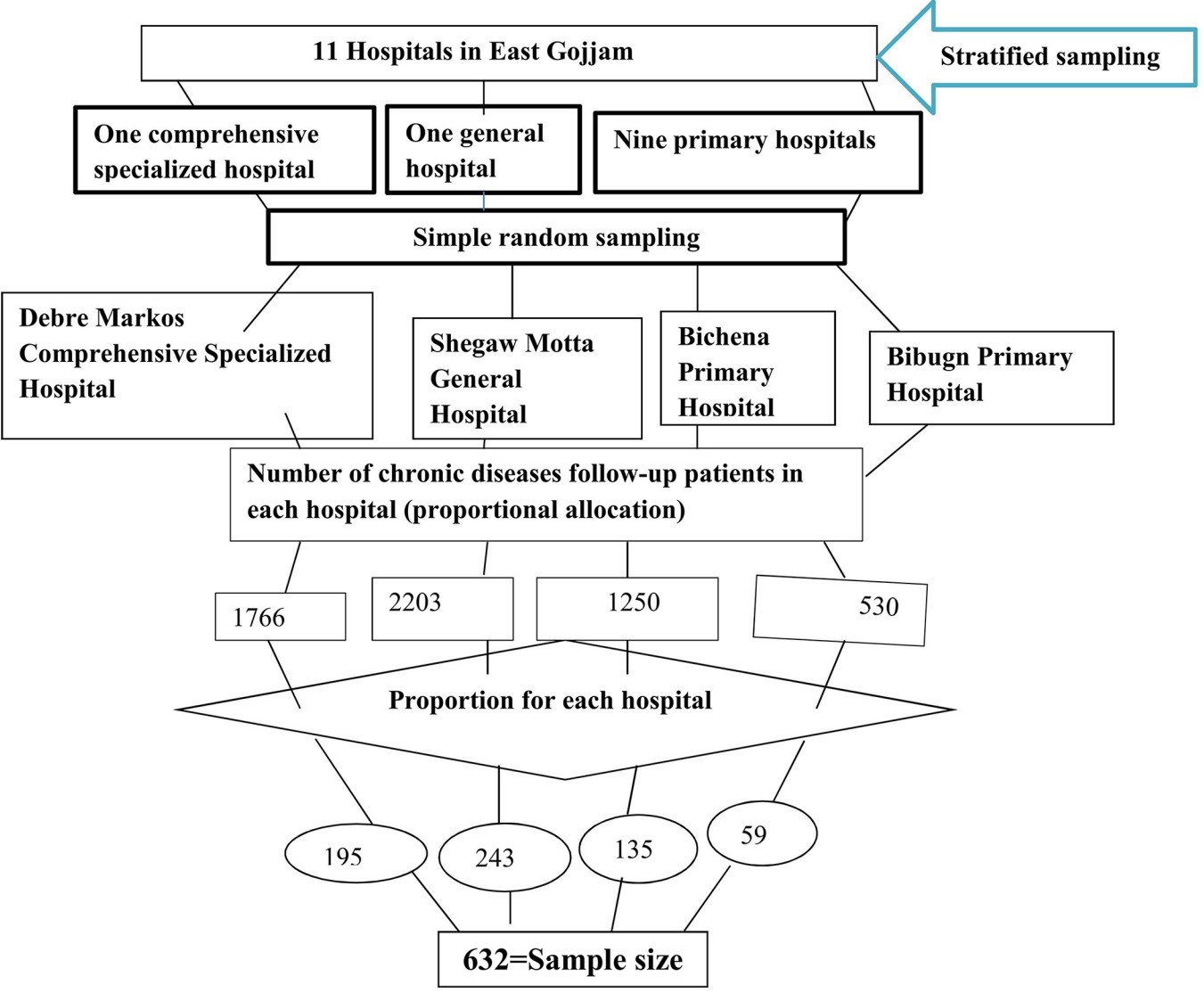

**Fig 1. Schematic presentation of sampling procedures for chronic patients enrolled in CBHI schemes.**

## Operational definitions

Seven scale items were used to determine how knowledgeable people were about CBHIS benefit packages. A yes/no choice of 1 or 0 was available for each item. Clients who correctly identified four or more CBHI benefit packages were considered to have good knowledge; otherwise, they lacked it [15]. On a five-point Likert scale, from strongly disagree to strongly agree, nine items related to CBHI satisfaction (CBHI offices opening times, information provision about CBHI scheme, timing/time interval of premium payment is convenient for your household, the collection process of insurance cards, recommend others to be a member of CBHI, CBHI benefit package meets the requirements of your household, the local CBHI management is trustworthy, the time to make use of the CBHI program after payment of registration fee, and need to stay enrolled in the CBHI scheme) were used to calculate the overall level of CBHI client satisfaction [14, 15]. After adding all the responses for each scale item, dichotomizing the overall level of satisfaction either satisfied (total satisfaction ≥75% value) or not satisfied (total

satisfaction<75% value) was made [13]. Chronic patients are those who have one or more chronic diseases, such as hypertension, diabetes mellitus, rheumatoid arthritis, epilepsy, heart failure, bronchial asthma, and others, according to a physician's diagnosis and documentation.

## Data collection tool and quality control

Data for the quantitative portion were gathered using an interviewer-administered structured questionnaire that was customized from several types of literature [13, 16, 22, 23] (**Fig 2**). The questionnaire asks about sociodemographic data, knowledge of the CBHI benefit package, CBHI experience, process and management-related issues, and health service-related aspects. To evaluate the questionnaire's clarity, consistency, and understandability, it was pretested on 5% (32 clients) of CBHI enrollees at Dejen Primary Hospital. Using Cronbach's alpha, the reliability of the tool for measuring items was 0.71. The Amharic version (the local language) was developed and then back-translated to the English version to maintain consistency. Six BSC nurses for collecting the data, and four health officers for supervision were recruited and trained. First, quantitative data was collected over three months (from February 28, 2022 to

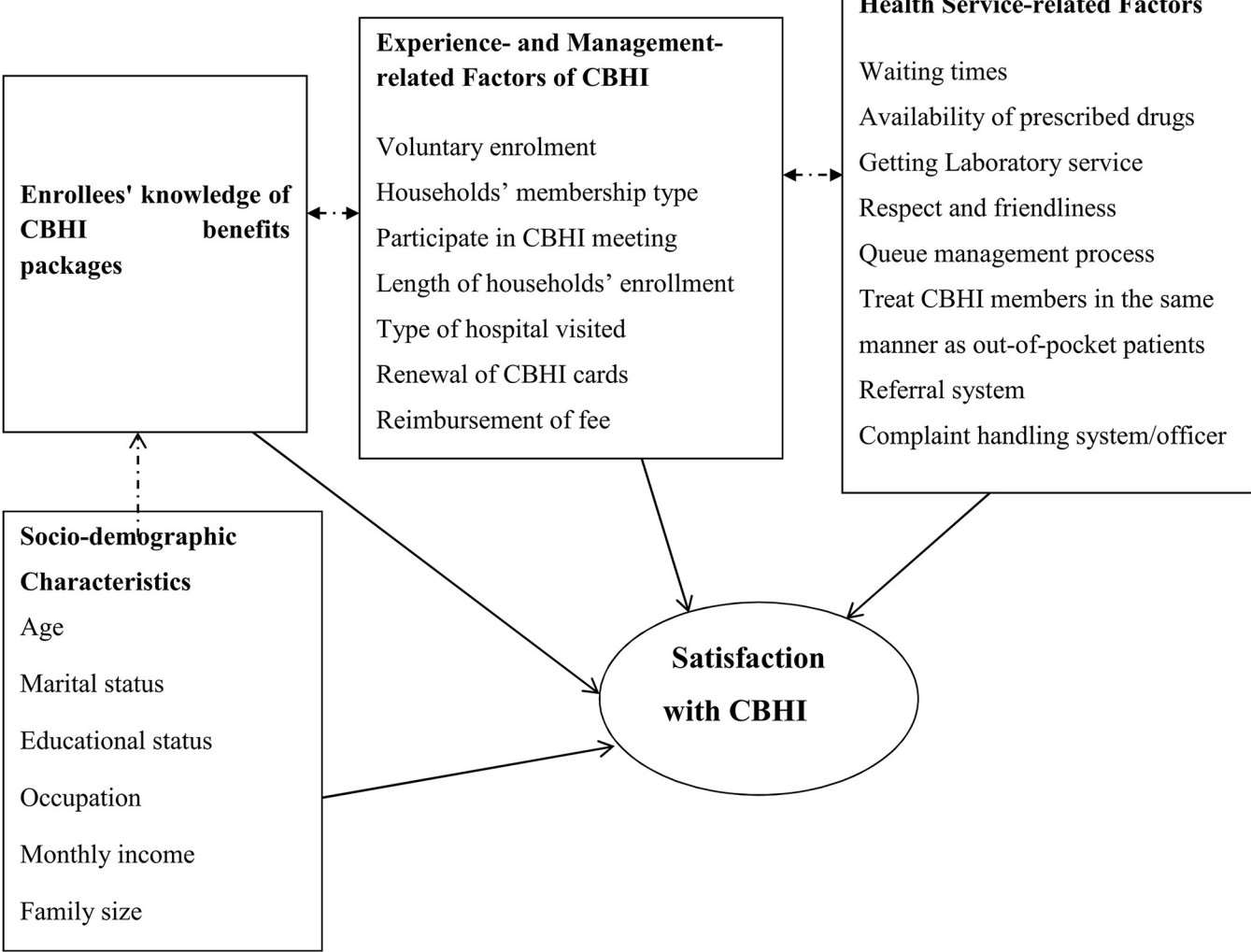

**Fig 2. Conceptual framework showing sociodemographic characteristics and other factors related to satisfaction with CBHI schemes.**

May 31, 2022), followed by qualitative data. An interview guide was used during key informant interviews. The interview was conducted in Amharic and was captured on a tablet recorder. Field notes were additionally recorded. Amharic-language transcription of the audio was done. After that, the information was translated into English. Finally, the findings incorporated participant direct quotes.

## Data processing and statistical analysis

Once the data had been cleaned, coded, and put into Epi-Data version 3.1, they were exported for analysis to SPSS version 25. Using descriptive statistics like frequency, percentage, mean, median, and interquartile range, the data were compiled and tabulated. For categorical variables, frequencies and percentages were reported; for continuous variables with normally distributed data, mean; and for data with a non-normal distribution, median and interquartile range. The multi-co-linearity test was checked to see the correlation between independent variables. To determine factors associated with the level of chronic patient satisfaction with the CBHI schemes, a binary logistic regression model was applied. The final model of multivariable analysis includes all variables with a p-value of less than 0.25 in bivariate analyses. Akaike information criteria and Bayesian information criteria were used to select the best model. Lemeshow's test to evaluate the goodness of fit, and the omnibus test to assess model fitness were used. We identified the factors using the adjusted odds ratio (AOR) and 95% confidence interval (CI), statistical significance at p-value 0.05 in multivariable binary logistic regression. The qualitative data transcripts were coded, followed by thematic analysis manually after reading and rereading the data. Both quantitative and qualitative data were separately analyzed and then integrated through a narrative approach.

## Results

### Socio-demographic characteristics

The study involved 632 chronic patients who completed structured questionnaires, with a response rate of 100%. Patient ages ranged from 25 to 82, with a mean age of 46.1 years (46.1± 5.2). A majority of the patients (60.6%) were between the ages of 45 and 64. The majority of patients (58.2%) lack a formal education. Farmers made up over a third (38.5%) of all study participants. Rheumatoid arthritis, hypertension, and diabetes mellitus were the three most common chronic diseases, respectively accounting for 36.4%, 24%, and 22% of all chronic patients. The remaining people were either affected by heart disease (9%), bronchial asthma (6%), or epilepsy (3%). More than half (59%) of chronic patients who use CBHI have incomes of less than 1000 ETB per month. The average monthly income for chronic patients was 1500 ETB (1500± 250). More than two-thirds of patients (68%) had families of five or more (**Table 1**).

### Knowledge of CBHI Benefit packages among chronic patients

More than half (56.2%) of household heads with chronic diseases knew the CBHI benefit packages. Nearly all of the participants agreed that CBHI was a good strategy for helping clients with healthcare costs (**Table 2**).

### Chronic patients' experience within CBHI schemes

Most (38.4%) had been to general hospitals. The majority of patients (91.5%) had enrollment periods longer than 12 months. More than two-thirds of them (79.3%) joined the CBHI voluntarily as small numbers (6.2%) would not renew their CBHI cards for the subsequent years.

**Table 1. Socio-demographic characteristics of chronic patients in the hospitals of East Gojjam, Northwest Ethiopia, 2022 (n = 632).**

| Variables | Category | Frequency | Percentage |
|---|---|---|---|
| Age | 25–44 | 136 | 21.5 |
| | 45–64 | 383 | 60.6 |
| | ≥65 | 113 | 17.9 |
| Sex | Male | 326 | 51.6 |
| | Female | 306 | 48.4 |
| Marital status | Married | 471 | 74.5 |
| | Single | 40 | 6.3 |
| | Widowed | 56 | 8.9 |
| | Divorced | 65 | 10.3 |
| Educational status | Not formal education | 368 | 58.2 |
| | Grade 1–8 | 155 | 24.5 |
| | Grade 9 and above | 109 | 17.2 |
| Occupation | Farmer | 244 | 38.6 |
| | Merchant and others* | 388 | 61.4 |
| Religion | Orthodox | 415 | 65.7 |
| | Muslim | 211 | 33.4 |
| | Protestant | 6 | 0.9 |
| Monthly income | ≤1000 | 372 | 58.9 |
| | 1000–5000 | 240 | 38 |
| | ≥5001 | 20 | 3.2 |
| Family size | 1–5 | 202 | 32 |
| | >5 | 430 | 68 |

Others

*include students, housewives, daily laborers

**Table 2. Knowledge of CBHI benefit packages among chronic patients (n = 632).**

| Variables | Category | Frequency | Percentage |
|---|---|---|---|
| A good strategy to help clients with health expenditure | Yes | 628 | 99.4 |
| | No | 4 | 0.6 |
| Covers only care from public health institutions | Yes | 590 | 93.4 |
| | No | 42 | 6.6 |
| Covers only care within the country | Yes | 503 | 79.6 |
| | No | 129 | 20.4 |
| Doesn't cover the transportation fee | Yes | 614 | 97.2 |
| | No | 18 | 2.8 |
| Covers outpatient care | Yes | 616 | 97.5 |
| | No | 16 | 2.5 |
| Covers inpatient care | Yes | 533 | 84.3 |
| | No | 99 | 15.7 |
| Doesn't cover medical care for cosmetic values | Yes | 394 | 62.3 |
| | No | 238 | 37.7 |

**Table 3. Chronic patients' experience within CBHI schemes (n = 632).**

| Variables | Category | Frequency | Percentage |
|---|---|---|---|
| Voluntary enrollment | Yes | 501 | 79.3 |
| | No | 131 | 20.7 |
| Renew CBHI membership | Yes | 593 | 93.8 |
| | No | 39 | 6.2 |
| CBHI related meeting | Yes | 208 | 32.9 |
| | No | 424 | 67.1 |
| Length of enrollment | <12 month | 54 | 8.5 |
| | >12 month | 578 | 91.5 |
| Household membership type | Paying members | 575 | 91 |
| | Subsidized members | 57 | 9 |
| Types of healthcare facilities accessed | Primary Hospital | 194 | 30.7 |
| | General Hospital | 243 | 38.4 |
| | Comprehensive Specialized Hospital | 195 | 30.9 |
| Reimbursement of fee | None/ Partial | 616 | 97.5 |
| | Fully | 16 | 2.5 |

Due to stock out, nearly all of the participants (97.5%) were unreimbursed or partially reimbursed customers who purchased from a private pharmacy or clinic (**Table 3**).

## Health service provision-related characteristics of chronic patients enrolled in the CBHI schemes

The majority of participants (65.5%) were kept waiting for longer than 50 minutes. The median wait time (interquartile range: 25–65 minutes) to see a physician was 53 minutes. Likewise, 66.1% of chronic patients receive respect and friendliness from healthcare providers as well as receiving the same level of out-of-pocket care (65.2%). A large percentage of chronic patients (94.6%) were unable to receive their prescribed drugs in hospitals, either partially or entirely (**Table 4**).

**Table 4. Health service provision-related characteristics of chronic patients enrolled in the CBHI schemes (n = 632).**

| Variables | Category | Frequency | Percentage |
|---|---|---|---|
| Waiting times to see the physician | >50 minutes | 414 | 65.5 |
| | ≤50 minutes | 218 | 34.5 |
| Being treated as CBHI members in the same manner as out-of-pocket patients | Yes | 412 | 65.2 |
| | No | 220 | 34.8 |
| Respect and friendliness | Yes | 418 | 66.1 |
| | No | 214 | 33.9 |
| Queue process | Yes | 218 | 34.5 |
| | No | 414 | 65.5 |
| Laboratory service provision | Yes | 335 | 53 |
| | No | 297 | 47 |
| Easy and fast referral pathway | Yes | 446 | 70.6 |
| | No | 186 | 29.4 |
| System for handling complaints | Yes | 164 | 25.9 |
| | No | 468 | 74.1 |
| Availability of drugs | None/ Partial | 598 | 94.6 |
| | Full | 34 | 5.4 |

## Satisfaction level of chronic patients with CBHI schemes

The overall satisfaction score for each participant was calculated, and using SPSS, a value of 75% was produced. About 196 (31%) of respondents scored at or above the level of satisfaction, whereas 436 (69%) did not.

## Factors affecting satisfaction with the CBHI schemes

At the bivariable binary logistic regression, the majority of variables from CBHI experience and hospital-related characteristics were significantly associated with chronic patients' satisfaction with the CBHI schemes. Finally, variables with p-values less than 0.25 were chosen for multi-variable binary logistic regression analysis. The multi-co-linearity test was checked using the variance inflation factor (mean VIF = 1.2). The model with the lowest Akaike information criterion and the Bayesian information criterion (AIC = 499.23, BIC = 488.5) could select eleven variables. Hosmer-Lemeshow's test, which was used to evaluate the goodness of fit test, was insignificant (p = 0.19), while the omnibus test, which assessed model fitness, had a p-value<0.001. Factors such as waiting times to see a physician, the respect and friendliness of healthcare providers, the availability of laboratory services, the availability of drugs, and the participant's knowledge were statistically significant at a p-value $\leq$ 0.05 at a 95% confidence interval. There was a decrease in patient satisfaction by 79% (AOR = 0.21; 95% CI: 0.07–0.61) and 76% (AOR = 0.24; 95% CI: 0.09–0.64), respectively, compared to full availability when drug availability was considered to be either nonexistent or partial. Chronic patients who were respected and treated with friendliness by care providers were seven times more likely to report being satisfied than those who were not (AOR = 7.045; 95% CI: 3.71–13.36). In comparison to chronic patients who were delayed for more than fifty minutes, those who received service within fifty minutes were almost twice as likely to be satisfied (AOR = 1.84; 95% CI: 1.12–3.00). Furthermore, the likelihood that chronic patients would be satisfied with the CBHI scheme was 1.6 times higher for those who used laboratory services compared to those who did not (AOR = 1.59; 95%CI:1.01–2.48). Similar to this, chronic patients with good knowledge were twice as likely to be satisfied as those with poor knowledge (AOR = 2.02; 95% CI: 1.24–3.27) (Table 5).

## Key informant interviews and thematic analysis findings

Twelve key informant in-depth interviewees (KIII), three from each institution, were recruited to explain the quantitative study's findings. The interviews with patients and service providers varied between 20 and 50 minutes. There were 16 sub-themes, 5 organizing themes, and 1 global theme constructed. Concerns with inputs and supply, service providers, finances, structural problems, and service users were the five organizing themes that emerged from the thematic analysis (Fig 3).

## Supply and inputs problems

This issue was pointed out as the main one in almost every in-depth interview with a key informant. The unavailability of drugs was essentially the primary concern that was raised by 11 key informants (two healthcare providers, four CBHI Executive Officers, and five chronic follow-up patients). One CBHI Executive officer, one health care provider (the medical doctor), and the third and five chronic follow-up patients also expressed discontent by patients with the lack of laboratory services. One CBHI Executive Officer and one chronic follow-up patient both quoted the lack of a backup community pharmacy.

**Table 5. Factors associated with satisfaction of chronic patients with the CBHI schemes in East Gojjam zone hospitals, Northwest Ethiopia (n = 632).**

| Variables | Client satisfaction | | AOR(95% CI) |
|---|---|---|---|
| | Yes | No | |
| **Age of the participant** | | | |
| 25–44 | 61(31.1) | 75(17.2) | 1.08(0.63–1.85) |
| 45–64 | 90(45.9) | 293(67.2) | 1.46(0.76–2.80) |
| ≥65 | 45(23) | 68(15.6) | 1 |
| **Monthly family income** | | | |
| ≤1000 | 125(63.8) | 247(56.7) | 0.93(0.56–1.55) |
| 1000–5000 | 65(33.2) | 175(40.4) | 0.91(0.28–2.98) |
| ≥5001 | 6(3.1) | 14(3.2) | 1 |
| **Participation at the meeting** | | | |
| Yes | 49(7.8) | 159(25.2) | 1 |
| No | 147(23.3) | 277(43.8) | 0.88(0.52–1.48) |
| **Household membership type** | | | |
| Paying | 172(27.2) | 403(63.8) | 1 |
| Subsidized | 24(3.8) | 33(5.2) | 0.67(0.33–1.37) |
| **Waiting times to the physician** | | | |
| >50 Minute | 70(11.1) | 344(54.4) | 1 |
| ≤50 Minute | 126(19.9) | 92(14.6) | 1.84(1.12–3.0)* |
| **Respect and friendliness of caregivers** | | | |
| No | 14(2.2) | 200(31.6) | 1 |
| Yes | 182(28.8) | 236(37.3) | 7.05(3.71–13.36)** |
| **Laboratory and other essential diagnostic tests** | | | |
| No | 60(9.5) | 237(37.5) | 1 |
| Yes | 136(21.5) | 199(31.5) | 1.59(1.02–2.48)* |
| **Availability of drugs in the hospital** | | | |
| None | 26(4.1) | 84(13.3) | 0.21(0.07–0.61)* |
| Partial | 143(22.6) | 345(54.6) | 0.24(0.09–0.64)* |
| Full | 27(4.3) | 7(1.1) | 1 |
| **Knowledge of participants** | | | |
| Poor knowledge | 46(7.3) | 231(36.6) | 1 |
| Good knowledge | 150(23.7) | 205(32.4) | 2.02(1.24–3.27)* |
| **Complaint handling system (officer)** | | | |
| Yes | 78(12.3) | 86(13.6) | |
| No | 118(18.7) | 350(55.4) | 1.34(0.84–2.14) |
| **Renewal of cards** | ' | | |
| Yes | 189(29.9) | 404(63.9) | |
| No | 7(1.1) | 32(5.1) | 0.94(0.33–2.64) |

*p.value<0.05

** p.value<0.001

*"When there is a shortage of drugs, there is a tendency for them to whine constantly because their pockets are empty. For one month, we have a deal with the supplier. When we offer our services, we issue a 3-month contract. This money won't be reimbursed promptly because of two months. Due to our delayed return, we are ineligible for entry. When they run out, there will be a loss of health insurance."***[43, male, Hospital Hospital Chief Executive Officers***]*

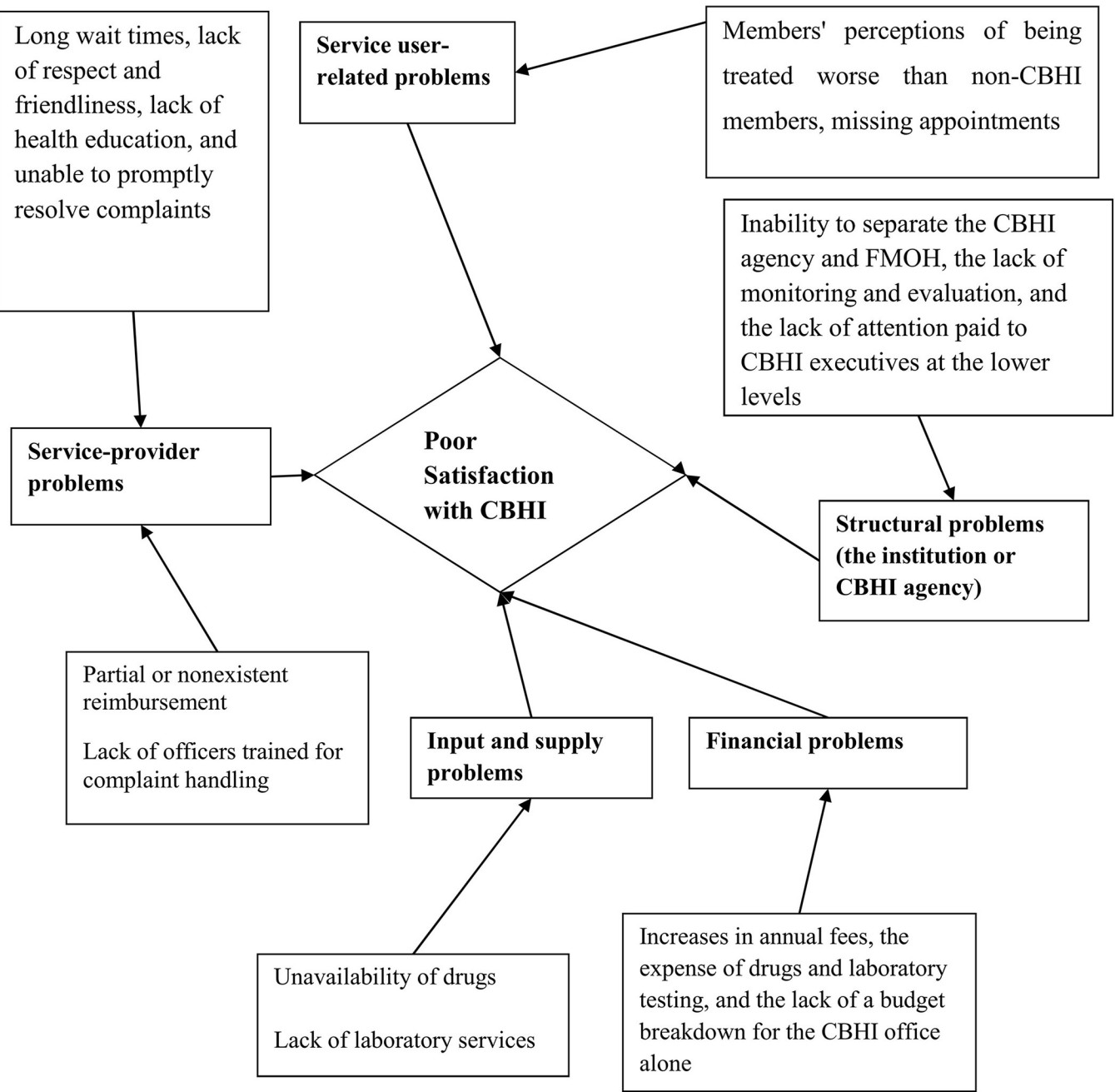

**Fig 3. Network of subthemes and main themes related to poor satisfaction of chronic patients with CBHI.**

*"Since we weren't able to get the laboratory service, we sent it outside. Thus, we are not required to pay. Some claim that paying is not worthwhile if we do not use it internally."*[**31, Male, CBHI executive officer**]

## Service provider problems

Most key informants mentioned service providers' issues such as long wait times, lack of respect and friendliness, lack of health education, and unable to promptly resolve complaints.

Problems with the service provider in the CBHI office included partial or nonexistent reimbursement and a lack of a professional trained for complaint handling.

Key informant interviews with one healthcare provider, two CBHI Executive officers, and two chronic follow-up patients highlighted that long wait times for chronic patients enrolled in the CBHI were an important concern.

*"I just got here this morning at six a.m., and I haven't had my turn yet. If there are any issues, I'm willing to stay up until lunch. These are quite, difficult, and upsetting for the sufferer. We skip the line. We have been operating for quite some time. Some people get disoriented and nod off while in line. A very dangerous situation exists right now."* [**45, Male, chronic patient**]

According to key informant in-depth interviewees (first, third, and fifth chronic follow-up patients, and fourth CBHI Executive Officers), the main challenge they encountered during their follow-up periods was the lack of respect and friendliness in hospitals, which contributed to their dissatisfaction with CBHI.

"*Being late can cause issues when interacting with a client. There is verbal abuse and bullying in the card room and nobody intervenes. These are all extremely serious issues. We enter the pharmacy and it continues in this manner. Like this, we seek out health insurance before purchasing any medications. And when we cope with it, it's only natural to feel sorry for ourselves. As far as we can tell, what we delivered was the sickness. Nobody wants to move quickly here.*" [**48, Male, chronic patient**]

**Financial issues:** Increases in annual fees, the cost of prescription drugs and laboratory testing at private pharmacies and clinics, and the absence of a budget breakdown for the CBHI office alone were among the financial issues that one healthcare provider (the nurse) and two CBHI Executive Officers addressed.

**Structural problems:** The Chief executive Officer, the health care provider (the doctor), and the first, second, and third CBHI Executive Officers all forwarded concerns about the failure to separate between the CBHI agency and the Federal Ministry of Health, the lack of monitoring and evaluation, and the disregard for CBHI executives at lower levels.

**Service-user problems**: The health care provider (the nurse) and one CBHI Executive Officer discussed the members' perceptions of being treated worse than non-CBHI members and their missing appointments.

## Discussion

Due to the emergent burden of chronic diseases, healthcare systems in many nations are failing to meet the demands of patients with these conditions, particularly as these countries adopt CBHI. Following the launch of Ethiopia's CBHI schemes, our study presented the first data concerning the satisfaction of CBHI by patients with chronic illnesses. The purpose of the current study was to determine the level of satisfaction with care provided to chronic patients, who were the beneficiaries of CBHI, to identify the factors that led to their satisfaction, as well as to explore the perspectives of healthcare providers at East Gojjam Hospitals in Northwest Ethiopia. Given that chronic patients were heads of household; our findings were compared using the previous studies that addressed how satisfied household heads were with the CBHI schemes. On top of that, there was a lack of comparable local data on satisfaction among chronic patients with CBHI.

The overall level of satisfaction was lower in our study when compared to those from China and Ethiopia's Boru Meda Hospital, Anilemo District, Sheko District, and Damote Woydie District [13–16, 24]. This may be explained by the fact that our study focused on chronic patients from households where the CBHI agreement might not cover the majority of the cost of drugs and laboratory testing, in contrast to previous studies that, except the qualitative study in China, focused on household heads. The other culprit can be the number of measurement components. As opposed to the five, six, and ten parameters in the prior studies, the current study included nine measurement items. Other considerations may encompass study setting (all studies, except for Boru Meda and Arsi, were community-based), socioeconomic factors, variations in study duration, and annual payment contributions (caused by yearly price increases). However, this very low level of patient satisfaction was similar to that of the qualitative study conducted at Bahirdar in Ethiopia [9] and was primarily attributed to the behavior of staff members, lengthy waiting times, the difficulties in obtaining medications and laboratory services, and the lack of education and information.

The satisfaction of the participants was inversely associated with the partial or non-availability of drugs. This is consistent with studies carried out across Ethiopia and Canada [9, 13, 14, 22, 25–27]. This could be attributed to the demand that patients pay additional fees to private pharmacies to obtain the drugs that were prescribed by the hospital. Patients' satisfaction with the CBHI schemes may have been reduced because these fees may not have been reimbursed. This was corroborated by almost all of the key informants.

Similarly, receiving easily available laboratory services had a significant effect on how satisfied chronic patients were with the CBHI schemes. This finding is in accordance with studies from Ethiopia, Bangladesh, and Canada [9, 14, 16, 25–30]. This is because many of the laboratory services covered by CBHI are frequently unavailable upon request, and patients are advised to find them elsewhere (at private clinics). The stock-out of laboratory services was cited as the primary challenge, according to interviews from care provider sides.

In accordance with the studies done in Nigeria, as well as Ethiopia's districts of Sheko, Anilemo, and Hadiya zone [14, 15, 18, 31], a higher level of satisfaction was found in study participants who had good knowledge about CBHI benefit packages than by those with poor knowledge. Chronic patients may be satisfied when they are aware of the CBHI benefit packages and how the health insurance system operates. This was supported by other studies in Ethiopia, Senegal, Eastern Sudan, and China [15, 17, 18, 24]. Lack of education and information related to CBHI was the most important factor identified from the key informant interview and it is consistent with the previous qualitative studies in Ethiopia and Canada [9, 25].

More importantly, patient satisfaction with the schemes was strongly associated with care provider respect and friendliness, with chronic patients who experienced higher levels of these qualities reporting higher levels of satisfaction. This is in harmony with the studies across Ghana and Ethiopia [17, 19, 20, 32–34]. This may be attributed to the fact that the motivated, competent, and compassionate (MCC) model has some applicability gaps. Most service providers and patients in the interview stated that many patients perceived a lack of respect and friendliness, which added to their dissatisfaction.

Finally, our study found that waiting times to a physician were strongly associated with satisfaction, with patients who waited less than 50 minutes being more satisfied than those who delayed for a long time. This affirms the findings of the studies undertaken in Ethiopia [13, 22, 25]. This could be ascribed to the study participants' chronic illnesses, which increase the likelihood that extended standing or sitting will make them irritated, annoyed, and dissatisfied. In-depth interviews also proved that long waits were the main reason of dissatisfaction to the extent that people began to feel disoriented and confused even when merely waiting in the queue.

Our study's strength is its use of mixed techniques; the findings of the quantitative study were strengthened and more explained by key information interviews. Along with data from CBHI beneficiaries, it attempted to investigate viewpoints from CBHI service providers. It provided the first data regarding the satisfaction of chronic patients with CBHI in our setting. Wider study areas at the zonal and different hospital levels were covered. Recall bias was reduced by using caregivers and chronic patients who received monthly check-ups. The lack of similar studies in Ethiopia and in Africa at large for comparison was the studies' major limitation. In addition, chronic patients who were not household heads were excluded from the study. It was impossible to evaluate the wealth quintile index for chronic patients. There were challenges to data due to the geriatric nature of chronic disease follow-up patients. We were unable to address crucial factors such as the attitudes of healthcare providers about insurance scheme membership, patient trust in contracted healthcare facilities, expectations and perceptions of the CBHI benefit, and the quality of healthcare services.

## Conclusion and recommendations

The overall CBHI scheme satisfaction was very low among patients seeking follow-up care for chronic diseases. Factors affecting the level of satisfaction were the respect and friendliness of service providers, waiting times, the availability of laboratory tests and drugs, and participant knowledge. The most dissatisfying aspect, which was substantiated by the qualitative findings, was the partial or non-availability of drugs for chronic diseases. Therefore, stakeholders must emphasize minimizing waiting times; improving the availability of drugs and laboratory tests at each hospital, opening back up community pharmacies, and promoting awareness of benefit packages through health education. Our study findings have significant policy-relevant implications for people with chronic disease. Future researchers should focus on nationwide studies covering more important factors.

## Acknowledgments

We would like to express sincere thanks to the staff members of the hospitals in the East Gojjam Zone, the data collectors, including the study participants for their vital participation in our study.

## Author Contributions

**Conceptualization:** Desalew Birhan, Zewudie Aderaw, Eskeziaw Agdew, Melkamu Siferih.

**Data curation:** Desalew Birhan, Eskeziaw Agdew.

**Formal analysis:** Desalew Birhan, Zewudie Aderaw, Eskeziaw Agdew, Melkamu Siferih.

**Investigation:** Desalew Birhan, Zewudie Aderaw, Melkamu Siferih.

**Methodology:** Zewudie Aderaw, Eskeziaw Agdew, Melkamu Siferih.

**Project administration:** Desalew Birhan, Zewudie Aderaw, Eskeziaw Agdew, Melkamu Siferih.

**Resources:** Desalew Birhan, Zewudie Aderaw, Eskeziaw Agdew, Melkamu Siferih.

**Software:** Desalew Birhan, Zewudie Aderaw, Eskeziaw Agdew, Melkamu Siferih.

**Supervision:** Desalew Birhan, Zewudie Aderaw, Eskeziaw Agdew, Melkamu Siferih.

**Validation:** Desalew Birhan, Zewudie Aderaw, Eskeziaw Agdew, Melkamu Siferih.

**Visualization:** Desalew Birhan, Zewudie Aderaw, Eskeziaw Agdew, Melkamu Siferih.

**Writing – original draft:** Desalew Birhan, Zewudie Aderaw, Eskeziaw Agdew, Melkamu Siferih.

**Writing – review & editing:** Desalew Birhan, Zewudie Aderaw, Eskeziaw Agdew, Melkamu Siferih.

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
