## [Decision Letter · Decision Letter 0]

24 Jul 2023

PGPH-D-23-01081

Satisfaction with Community-Based Health Insurance Scheme and Its Associated Factors among Chronic Disease Follow-up Patients in East Gojjam Zone Hospitals, Northwestern Ethiopia, 2022: Mixed Method Study

Dear Dr. Zeleke,

Thank you for submitting your manuscript to PLOS Global Public Health. After careful consideration, we feel that it has merit but does not fully meet PLOS Global Public Health’s publication criteria as it currently stands. Therefore, we invite you to submit a revised version of the manuscript that addresses the points raised during the review process.

We look forward to receiving your revised manuscript.

Kind regards,

Humayun Kabir

Academic Editor

Journal Requirements:

1. Please amend your online Financial Disclosure statement. If you did not receive any funding for this study, please simply state: “The authors received no specific funding for this work.”

2. Please update your online Competing Interests statement. If you have no competing interests to declare, please state: “The authors have declared that no competing interests exist.”

3. In the online submission form, you indicated that "SPSS data,the questionare and, other important data can be accessed from the main author upon reasonable request.". All PLOS journals now require all data underlying the findings described in their manuscript to be freely available to other researchers, either 1. In a public repository, 2. Within the manuscript itself, or 3. Uploaded as supplementary information.

4. Please provide separate figure files in .tif or .eps format only and ensure that all files are under our size limit of 10MB.

Additional Editor Comments (if provided):

Please write the exclusion inclusion criteria as a sentence, not by bullet points.

Same for the operational definition and sample size.

As the binary logistic regression was done, talk about the model assumption.

Report the AIC and BIC,

And provide the ROC curve and AUC.

Reviewers' comments:

Reviewer's Responses to Questions

**Comments to the Author**

1. Does this manuscript meet PLOS Global Public Health’s publication criteria? Is the manuscript technically sound, and do the data support the conclusions? The manuscript must describe methodologically and ethically rigorous research with conclusions that are appropriately drawn based on the data presented.

Reviewer #1: Yes

Reviewer #2: Yes

2. Has the statistical analysis been performed appropriately and rigorously?

Reviewer #1: No

Reviewer #2: Yes

3. Have the authors made all data underlying the findings in their manuscript fully available (please refer to the Data Availability Statement at the start of the manuscript PDF file)?

Reviewer #1: Yes

Reviewer #2: No

4. Is the manuscript presented in an intelligible fashion and written in standard English?

Reviewer #1: Yes

Reviewer #2: No

5. Review Comments to the Author

Reviewer #1: Thank you for inviting me to review this manuscript. Overall, the paper is good and I have put my comments on where the clarification is needed to consider the paper for publication.

• Abstract: the introduction, methods used, results, and conclusions were well summarized.

• Introduction: The authors tried to show the gaps in the previous study. However, the target population or study population of the previous studies that were cited in the manuscript was not chronic patients. The limitations of the previous study, which focused on chronic patients, could be the reason why the current study was needed. So, clearly stating the problems, showing what efforts have been made, what the gaps were, and how the current study fills the identified gaps should be clearly shown in the introduction section.

Methods and participants

• Sample size determination: The sample size was determined based on the previous study, ‘Magnitude of Satisfaction and Associated Factors Among Household Heads Who Visited Health Facilities with Community-Based Health Insurance Scheme in Anilemo District, Hadiya Zone, Southern Ethiopia," and its study population was households, but the current study focused on individual chronic patients who were visiting hospitals for chronic follow-up care. So, using a previous study with a different study population for sample determination is inappropriate. So, it's not clear why the authors used such a study.

• The method used was a mixed approach, which was sequential and explanatory. Here, clarity is important: what about sequential? Is it quan-qual or the reverse? Why was a mixed approach needed? Was there any aspect that could not be addressed either quantitatively or qualitatively?

• Study variables: I think some variables were missed. for instance, trust in contracted health facilities, health status, and the attitude of the health care providers towards insurance scheme membership. quality of health services, accessibility and availability of health services, health status,...

• The measurement of some study variables lacked clarity. For example, how were the waiting time (for those participants who were not able to read and write about waiting time while looking at the watch to respond to the interview), queue management process, and complaint handling system measured?

• The wealth index is more appropriate than income in an insurance scheme. Why did the authors prefer monthly income instead of the wealth index? Because the current members of the CBHI scheme are from the informal sector in Ethiopia, they do not have a monthly income.

• Data collection: For quantative data collection, the authors have mentioned that open questions were used. In addition, the key informant interview was used for the qualitative part. For what type of variables were open-ended questions used, and what was their importance in quantitative data?

• Results: The authors described the total number of chronic disease follow-up patients. But here, only describing the determined sample size, n = 632, was more relevant.

• It was mentioned, ‘Then, chronic follow-up clients were classified as satisfied (if they scored a 32 satisfaction score) and not satisfied (if they scored below a 32 satisfaction score)’. But it was operationalized as "satisfied CBHI client: the overall level of satisfaction greater than or equal to seventy-five percent response rate for each scale item otherwise not satisfied CBHI client’. So, it lacked consistency.

• Table titles: It would be more appropriate if the ‘clients’ were replaced with "chronic disease follow-up patients" to make it more clear.

• In Table 5, the aim of bivariable and multivariable binary logistic regression analyses was to identify the associated factors. So, the title of the table should be modified to’ Factors Affecting Satisfaction towards the CBHI Scheme among Chronic Diseases: Follow-up Patients."

• In Table 5, religion was considered an associated factor in bivariable logistic regression, which showed the odds of satisfaction with the CBHI scheme among orthodox were three times higher compared to protestants. In addition, the number of observations satisfied was only 1, which was less than 5 in a cell. So, what will the authors recommend? please recheck.

• In table 5, in the column of client satisfaction, the response categories (satisfied and dissatisfied) were not shown.

Discussion and Conclusions

• The results were repeated rather than discussed in this section, and the results were discussed compared to similar study findings, but the study population was quite different.

• Strength: I could not see the special strength of the study. Using qualitative methods could not make the study so strong.

• Conclusions: The result was concluded as follows: "The overall proportion of satisfaction in chronic disease follow-up patients in this study was 31%’. It would be appropriate if it were concluded using appropriate terms like lower, moderate, or higher regarding the level of satisfaction rather than a number or percentage.

•In conclusion section, instead of using ‘respondent," simply name the study participants for whom the study findings are concluded.

•Conceptual Framework: The variable, family size was shown in the conceptual framework but not listed in the variable lists. Similar to this, some variables were listed in the study variables but were not included in the conceptual framework. For instance, waiting time.

Reviewer #2: The manuscript has merit . it is fairly presented in an intelligent fashion however it lacks standard english. there is no evidence of data availability. other comments to be addressed are attached.

6. PLOS authors have the option to publish the peer review history of their article (what does this mean?). If published, this will include your full peer review and any attached files.

**Do you want your identity to be public for this peer review?** For information about this choice, including consent withdrawal, please see our Privacy Policy.

Reviewer #1: **Yes: **Edosa Tesfaye Geta

Reviewer #2: **Yes: **DR ESTER LILIAN ACEN

<quillbot-extension-portal></quillbot-extension-portal>

---

## [Decision Letter · Decision Letter 1]

13 Nov 2023

PGPH-D-23-01081R1

Satisfaction of Chronic Patients with Community-Based Health Insurance Schemes and Related Factors: Explanatory Sequential Mixed Methods Study

Dear Dr. Zeleke,

Thank you for submitting your manuscript to PLOS Global Public Health. After careful consideration, we feel that it has merit but does not fully meet PLOS Global Public Health’s publication criteria as it currently stands. Therefore, we invite you to submit a revised version of the manuscript that addresses the points raised during the review process.

We look forward to receiving your revised manuscript.

Kind regards,

Humayun Kabir

Academic Editor

Journal Requirements:

Additional Editor Comments (if provided):

Reviewers' comments:

Reviewer's Responses to Questions

**Comments to the Author**

1. If the authors have adequately addressed your comments raised in a previous round of review and you feel that this manuscript is now acceptable for publication, you may indicate that here to bypass the “Comments to the Author” section, enter your conflict of interest statement in the “Confidential to Editor” section, and submit your "Accept" recommendation.

Reviewer #1: (No Response)

Reviewer #2: All comments have been addressed

2. Does this manuscript meet PLOS Global Public Health’s publication criteria? Is the manuscript technically sound, and do the data support the conclusions? The manuscript must describe methodologically and ethically rigorous research with conclusions that are appropriately drawn based on the data presented.

Reviewer #1: Partly

Reviewer #2: Yes

3. Has the statistical analysis been performed appropriately and rigorously?

Reviewer #1: Yes

Reviewer #2: Yes

4. Have the authors made all data underlying the findings in their manuscript fully available (please refer to the Data Availability Statement at the start of the manuscript PDF file)?

Reviewer #1: Yes

Reviewer #2: Yes

5. Is the manuscript presented in an intelligible fashion and written in standard English?

Reviewer #1: (No Response)

Reviewer #2: Yes

6. Review Comments to the Author

Reviewer #1: Thank you for revising the manuscript. The authors have incorporated some comments and provided explanations for why they did not incorporate most of the comments, but the explanations are not scientifically sound and convincing. Regarding the revised manuscript, my major concern that has not been addressed yet is the reanalysis of the data.

Based on the previous comments, in Table 5, the authors explained that some variables were removed during reanalysis, i.e., 11 variables were presented in Table 5 of the revised manuscript and 17 variables were presented in Table 5 of the first draft manuscript. About six variables were removed. However, none of the AOR values were changed for all variables. When variables are added or removed from the analysis, the value of the analysis outputs (AOR) is expected to be changed, but this has not happened. This shows that the authors did not reanalyze the data; they simply removed the variables from the tables. This is a big mistake, and I doubt the analysis results presented in the revised manuscript were not from relevant candidate variables.

The authors did not possess data suitable for answering their research question and my concerns. Therefore, I suggest the paper should be rejected.

Reviewer #2: most of the comments have been adressed

7. PLOS authors have the option to publish the peer review history of their article (what does this mean?). If published, this will include your full peer review and any attached files.

**Do you want your identity to be public for this peer review?** For information about this choice, including consent withdrawal, please see our Privacy Policy.

Reviewer #1: No

Reviewer #2: **Yes: **Ester Lilian Acen

---

## [Editor Report · Decision Letter 2]

28 May 2024

Satisfaction of Chronic Patients with Community-Based Health Insurance Schemes and Related Factors: Explanatory Sequential Mixed Methods Study

PGPH-D-23-01081R2

Dear Melkamu,

We are pleased to inform you that your manuscript 'Satisfaction of Chronic Patients with Community-Based Health Insurance Schemes and Related Factors: Explanatory Sequential Mixed Methods Study' has been provisionally accepted for publication in PLOS Global Public Health.

Best regards,

Humayun Kabir

Academic Editor